# Characteristics Associated with Burnout among Cardiologists in an Academic Medical Setting: Baseline Survey Results from a Communication Coaching RCT

**DOI:** 10.3390/bs12100362

**Published:** 2022-09-27

**Authors:** Travia Kayla Dunbar, Maren K. Olsen, Hongqiu Yang, Danielle Kennedy, Larry R. Jackson, Kevin L. Thomas, Aviel Alkon, Neil S. Prose, Kathryn I. Pollak

**Affiliations:** 1Department of Population Health Sciences, Duke University School of Medicine, Durham, NC 27710, USA; 2Department of Biostatistics and Bioinformatics, Duke University School of Medicine, Durham, NC 27710, USA; 3Durham Center of Innovation to Accelerate Discovery and Practice Transformation, Durham Veterans Affairs Health Care System, Durham, NC 27705, USA; 4Duke Cancer Research Institute, Durham, NC 27710, USA; 5Division of Cardiology, Duke University School of Medicine, Durham, NC 27710, USA; 6Duke Clinical Research Institute, Duke University School of Medicine, Durham, NC 27705, USA; 7Department of General Internal Medicine, Duke University School of Medicine, Durham, NC 27710, USA; 8Department of Dermatology and Pediatrics, Duke University School of Medicine, Durham, NC 27710, USA

**Keywords:** physician burnout, cardiology, Maslach Burnout Inventory, emotional exhaustion, communication study

## Abstract

Objectives: Clinician burnout poses risks not just to clinicians but also to patients and the health system. Cardiologists might be especially prone to burnout due to performing high-risk procedures, having to discuss serious news, and treating diseases that incur significant morbidity and mortality. Few have attempted to examine which cardiologists might be at higher risk of burnout. Knowing at-risk cardiologists can help frame resilience interventions. Methods: We enrolled 41 cardiologists across five ambulatory cardiology clinics into a randomized controlled trial where we assessed the Maslach Burnout Inventory at baseline. We used bivariate analyses to assess associations between cardiologist demographics and burnout. Results: Cardiologists reported low burnout for depersonalization and personal accomplishment and moderate levels for emotional exhaustion. Female cardiologists reported emotional exhaustion scores in the “low” range (M = 12.3; SD = 10.06), compared to male cardiologists whose score was in the “moderate” range 19.6 (SD = 9.59; *p* = 0.113). Cardiologists who had greater than 15 years in practice reported higher mean scores of emotional exhaustion, indicating moderate burnout (M = 20.0, SD = 10.63), compared to those with less than 15 years in practice (M = 16.6, SD = 9.10; *p* = 0.271). Conclusions: In this sample, unlike prior studies, male cardiologists reported more burnout. Consistent with prior work, mid-level cardiologists might be at highest risk of emotional exhaustion.

## 1. Introduction

Providing high quality care demands much of clinicians. Clinician burnout has recently received much attention with estimates that almost half of physicians experience burnout, Ref. [1] and estimates higher after the COVID-19 pandemic [2,3]. The International Classification of Diseases (ICD-11) defines burnout as a syndrome resulting from chronic workplace stress not successfully managed [4]. Feelings of exhaustion, increased feelings of cynicism related to an individual’s job, and reduced efficacy are classified dimensions of burnout [4].

## 2. Literature Review

Burnout has many causes, including setting of practice, work–life balance, long hours, and electronic health records [5,6,7,8,9,10,11,12]. Some physicians might be more prone to burnout. Some have found the female physicians have higher burnout [5,13,14]. Another factor that can affect burnout is being a specialist vs. a primary care clinician [15]. One specialist group that has been understudied is cardiologists. Cardiologists provide inpatient and outpatient care, and some perform complex, life-altering procedures that include risks for patients. Further, cardiologists often deliver serious news, and at times, must deal with patients facing terminal illnesses. Although many have examined factors related to burnout in oncology [7], which is another specialty that treats serious illness, few have studied burnout in cardiology [1,2,15].

The one study that focused on cardiologists found that more than one quarter of U.S. cardiologists reported burnout [16]. However, the authors only included one component of burnout from the Maslach Burnout Inventory Scale [17], emotional exhaustion. Moreover, the authors had a low response rate (21% of 10,798 cardiologists approached), which limits generalizability. More work is needed to understand which cardiologists are at higher risk of burnout as health systems design resilience programs [18] and can target them. In this brief report, we examine demographic predictors of burnout assessed at baseline among cardiologists who participated in the Communication in Coaching in Cardiology (CCC) trial [19]. 

## 3. Methods

### 3.1. Participants

We recruited cardiologists starting in February 2019 and ending in March 2020 to participate in a communication study across 5 ambulatory clinics and obtained written consent. We attended faculty meetings to approach cardiologists in small group settings or clinic leaders sent out emails introducing the study. Seventy-eight clinicians met inclusion criteria of working at least a half day a week. We randomly selected 511 of the 78 cardiologists, and 41 consented (8181%). All 41 cardiologists completed the baseline survey (100% completion rate).

### 3.2. Baseline Survey Measures

We assessed demographics, clinical factors, and psychosocial factors. Demographic factors included age, sex, and race. Clinical factors included average amount of patient care hours and years in practice. Psychosocial factors included strength in religious beliefs. 

### 3.3. Primary Outcome: Burnout

We assessed burnout using the abbreviated Maslach Burnout Inventory (MBI) [17]. The MBI has three components: emotional exhaustion, depersonalization, and personal achievement. Emotional exhaustion includes chronic fatigue and physical problems outside of the workplace. Low burnout is classified with a score <17, moderate is 18–29, and high is ≥30. Depersonalization refers to dehumanization or detachment from patients. Low burnout is <5, moderate is 6–11, and high is ≥12. Personal achievement involves feeling enriched by work and feeling a sense of accomplishment and joy. Low burnout is ≥40, moderate is 34–39, and high is ≤33 [20]. 

## 4. Statistical Analyses

We used bivariate analyses to assess mean differences on the three burnout subscales based on cardiologist characteristics. We used analysis of variance (ANOVA) for multi-category characteristics or t-tests for two-category characteristics. Due to the limited sample size, we evaluated differences for both clinical significance and statistical significance at *p* < 0.05. We used SAS v 9.4 (Cary, NC, USA) for all analyses.

## 5. Results

Please see Table 1 for demographic characteristics of our sample. Although none of the associations achieved a statistical significance of *p* < 0.05, there were several notable clinically significant associations when examining the emotional exhaustion component of the MBI (Table 2). First, cardiologists in this sample reported low depersonalization (M = 5.15, SD = 4.29) and relatively high personal achievement (M = 40.28, SD = 4.845). However, their levels of emotional exhaustion fell in the moderate level (M = 18.28, SD = 10.05). Male cardiologists reported higher mean scores of emotional exhaustion, indicating moderate burnout (mean = 19.6, SD = 9.59), compared to females indicating low burnout (mean = 12.3, SD = 10.06) (*p* = 0.113). Cardiologists who had greater than 15 years in practice also reported higher mean scores of emotional exhausting, indicating moderate burnout (mean = 20.0, SD = 10.63), compared to those with less than 15 years in practice (mean = 16.6, SD = 9.10) (*p* = 0.271). The two other MBI subscales of depersonalization and personal accomplishment exhibited less variability; we found no differences in these based on cardiologist characteristics (See Table 2).

## 6. Discussion

This analysis had several notable trends. First, cardiologists in this sample did not exhibit much burnout, except for emotional exhaustion. Further, female cardiologists reported low emotional exhaustion scores while males reported moderate emotional exhaustion. In addition, cardiologists practicing for more than 15 years on average reported higher scores on all three subscales. Although these results did not reach statistical significance, the variability across means in each predictor deserves exploration. 

When reviewing the burnout literature, our findings confirm and disconfirm prior results. In the only prior study among cardiologists, the authors found that those who identified as female and were middle-aged experienced higher levels of burnout [16]. Being female has also been related to higher physician burnout physicians in general [2,13,14]. Thus, our findings showing that males reported moderate emotional exhaustion, compared females reporting low, differs from prior work. The lack of consistency may reflect our small sample size and the low number of females in our sample. Learning what might have helped females feel less burned out in our academic setting might help females in other specialties and in other practice settings.

Some of our results support prior work but others do not support prior work. In our analysis, we found that those practicing more than 15 years averaged higher levels of emotional exhaustion. This finding is consistent with the only study that examined cardiologists [16]. However, among other physicians, some have found that early career physicians struggle more [14]. It might be that early career cardiologists receive a lot of support, but mid-career cardiologists receive less and have more administrative and mentoring demands. Those developing resilience programs can possibly learn about what buffers early career cardiologists use to help those who transition to mid-career who might need more support. 

Lastly, the work on burnout is hampered by the variability in methods to assess burnout. In the systematic review conducted by Rotenstein et al., at least 47 definitions for burnout prevalence were identified [1]. Many investigators have not included the full burnout scale. Our analysis included the full scale of emotional exhaustion, personal accomplishment, and depersonalization; however, interestingly, we only found differences in emotional exhaustion, which might be the most remediable by resilience programs. Moreover, cardiologists in our sample had high personal achievement, which can represent good coping mechanisms to combat emotional exhaustion. Developing a definitive and standardized approach to burnout will assist in targeting those who may experience symptoms more often. 

These results should be interpreted with several noted limitations. The main limitation is that our sample was small. Further, we only included cardiologists in academic settings, which limits generalizability. We were also limited in the characteristics we assessed and might have missed important predictors. We were also unable to assess burnout profiles, instead using cutoffs, which might not have adequately portrayed burnout [21]. Finally, cardiologists were also enrolled in a communication study; those who had higher burnout might have refused the study.

## 7. Conclusions

Our results indicate that, in general, cardiologists in this small sample seemed to report resilience and only some burnout in terms of emotional exhaustion. However, some groups might benefit more from resilience training, particularly resilience trainings that focus on gender and years in practice. Many have developed effective trainings [18]. Future directions should explore these findings in a larger and more diverse sample to verify these potential differences.

### Practice Implications

With a growing demand for physicians, including cardiologists, understanding predictors of burnout is paramount for resilience interventions with a long-term goal of maintaining a diverse and productive workforce.

## Figures and Tables

**Table 1 behavsci-12-00362-t001:** Cardiologist demographic characteristics.

TC Table 4.0a Physician Demographics	Total (n = 41) N (% or M, SD)
**Gender (%)**	
Female	7 (17%)
Male	34 (83%)
**Age**	
Mean (SD)	47 (9)
Median (25th, 75th)	46 (41, 54)
**Race (%)**	
Asian	10 (24%)
Black or African American	2 (5%)
White	26 (63%)
Another race	3 (7%)
**Ethnicity (%)**	
Hispanic or Latino	3 (7%)
Not Hispanic or Latino	37 (90%)
Prefer not to answer	1 (2%)
** Direct patient hours per week**	
Mean (SD)	30 (14)
Median (25th, 75th)	30 (20, 40)
**Practice specialty (%)**	
General	14 (34%)
Electrophysiology	7 (17%)
Interventional	8 (20%)
Advanced Heart Failure	8 (20%)
Cardiac Imaging	3 (7%)
Congenital disease	1 (2%)
**Years in Practice (%)**	
Less than a Year	1 (2%)
1–5 Years	3 (7%)
6–10 Years	7 (17%)
11–15 Years	9 (22%)
More than 15 Years	21 (51%)
**Strength of Religious/Spiritual Beliefs (Range: 1–10)**	
Mean (SD)	5 (3)
Median (25th, 75th)	6 (3, 8)

**Table 2 behavsci-12-00362-t002:** Descriptive statistics of burnout scales by cardiologist characteristics *.

	N	Emotional Exhaustion	Depersonalization	Personal Accomplishment
		Mean (SD) Median, 25th, 75th	Mean (SD) Median, 25th, 75th	Mean (SD) Median, 25th, 75th
**Gender at birth**				
Male	34	19.6 (9.59) 18, 13, 25	5.2 (4.22) 5, 2, 8	39.9 (4.81) 41, 37, 44
Female	7	12.3 (10.06) 8, 5, 24	4.6 (4.79) 3, 1, 8	42.1 (4.49) 42, 38, 47
***p*-value**		0.113	0.764	0.267
**Race**				
White	26	18.8 (9.23) 19, 11, 25	5.2 (4.21) 5, 2, 7	40.7 (5.18) 42, 37, 45
Non-white	15	17.6 (11.38) 15, 10, 24	4.9 (4.49) 3, 1, 8	39.6 (4.07) 40, 36, 44
***p*-value**		0.729	0.8217	0.4604
**Direct patient care hours per week**				
0–30	24	18.1 (10.21) 17, 12, 23	5.1 (4.63) 4, 2, 7	39.5 (5.20) 41, 37, 44
31–60	17	18.8 (9.85) 20, 10, 25	5.0 (3.82) 5, 2, 8	41.4 (4.00) 42, 37, 45
***p*-value**		0.831	0.925	0.192
**Strength of religious/spiritual beliefs**				
Low beliefs (0–4)	14	18.6 (9.22) 18, 11, 25	4.8 (4.04) 4, 2, 6	39.1 (5.41) 41, 37, 42
Moderate beliefs (5–7)	16	18.7 (8.12) 18, 13, 25	5.6 (3.88) 5, 3, 8	40.6 (3.69) 41, 37, 44
Strong beliefs (8–10)	11	17.6 (13.61) 15, 8, 25	4.6 (5.30) 2, 1, 11	41.5 (5.41) 42, 37, 47
***p*-value**		0.962	0.808	0.457
**Years in practice (post-graduate training)**				
≤15 years	20	16.6 (9.10) 15, 10, 23	4.7 (4.33) 3, 2, 6	39.2 (4.26) 40, 37, 42
>15 years	21	20.0 (10.63) 21, 12, 25	5.5 (4.26) 5, 2, 8	41.3 (5.11) 42, 39, 45
***p*-value**		0.271	0.542	0.154

* *p*-values calculated from ANOVA or t-test, as appropriate. For emotional exhaustion: low (0–16), moderate (17–26), and high (≥27). For depersonalization: low (0–6), moderate (7–12), and high (≥13). For personal achievement: low (≤31), moderate (32–38), and high (≥39).

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
