# Peer review of "Characteristics Associated with Burnout among Cardiologists in an Academic Medical Setting: Baseline Survey Results from a Communication Coaching RCT"

_behavsci, 2022, doi:10.3390/bs12100362_

Round 1

Reviewer 1 Report

Dear Author, I was delighted to have gotten the opportunity to review your paper. I work with health professionals and we provide different trainings to improve their mental health. Thus, I was interested in your study. At the same time, I have quite a few comments, see in the attached file. 

Author Response

  1. Please indicate in the Title if it is a part of a study (baseline survey).

Thank you. We added this to the title.

  1. Bivariate analyses (it is the common term in statistics)

Thank you. We changed this.

  1. Please what does this difference mean like you did in the next sentence (“Female cardiologists reported a lower mean emotional exhaustion score…”)

We have clarified that females reported low emotional exhaustion and males reported moderate.

  1. At least, in the case of the emotional exhaustion (referring to Conclusion in Abstract). Please indicate this here

Thank you. We edited it and said emotional exhaustion.

  1. Question 1: The dispersion is relatively high. How did you manage this high variability in the data. (It is a bias/limitation, small sample size with a high variability).

Thank you. We present the confidence intervals given the small sample and wide variability.

  1. It is important to note that there are a lot of studies on burnout among health care professionals.

Yes. We have added extensively to the literature review. We thought we were limited in the number of citations we had, given this is a Brief Report.

  1. It is recommended to write this into a new paragraph and add a clear aim of the present study. If it is a part of a big study, it is recommended to provide the aim of the big study and the importance of the baseline survey. Question 2: Did you have any prior hypothesis?

Thank you. We clarify the purpose of the article. Given the variability in the literature, we did not have hypotheses.

  1. “future interventions” such as for example resilience trainings (?)

Yes. Thank you. We clarified this.

  1. In the abstract, the authors indicated that it is a two-arm cluster randomized controlled trial which not presented in the Methods section. It will be useful to give what does two-arm cluster-randomized controlled trial mean in the present study. What does this baseline survey present in this trial?

Thank you. We removed that as it is distracting to the purpose of this paper.

  1. There are different methods to classify people based on the scores of the burnout dimensions: emotional exhaustion, depersonalization and diminished personal accomplishment. Certain classification methods divide the normative population into three categories and used arbitrary cut-off scores for each, but these methods lack diagnostic validity (it is a limitation).

Please cite the used cut-off scores, by who? for example by Canouï, Mauranges?

Thank you. We cite our cutoffs.

  1. Using cut-off score is a little bit old-fashioned, Leiter & Machlach (2016) recommended to use burnout profiles instead of cut-off scores.

We agree. Unfortunately, we do not have the data to look at profiles. We now list this as a limitation.

  1. the statistical term is bivariate analyses.Or it is recommended: ANOVA and t-test were used for comparison of means between groups on burnout dimensions. The term association is related to test associations (for example: correlation, cross-tabs) not compare means.

Thank you. We have clarified this.

  1. It is rather the practical significance. How evaluate the clinical significance? It is recommended to use unbiased effect size measurements for ANOVA omega-squared and for t-tests Hedges' g.

Thank you. With our small sample, we are unable to do this.

  1. What were the scores in the burnout dimensions in the sample? What were the frequencies of burnout categories based on cut-off scores in the sample?

Thank you. We have added this.

  1. In sum, examining the burnout dimensions what is a typical profile among cardiologist? OK, sample size is very low but it would be useful to examine the profiles too to explore the burnout among cardiologists. It is also recommended to examine the difference in frequency distributions, are there any difference in the ratio of burnout categories (low, moderate, high) among the examined groups?

Thank you. We now list the profile for the entire sample to show the overall burnout. We also have noted that there was only one difference with females being low and males being moderate.

  1. Please delete the rows with 0 value (it is irrelevant). It is recommended to separate the different factors (demographic, clinical work, psychosocial). I missed the Confidence factor in Table 1 and Table 2.

We deleted those rows. We did not end up testing the confidence due to multiple testing and have removed it from the paper.

  1. M, SD - two decimal places; p - three decimal places. It is recommended based on APA, AMA guideline how to report results.

Thank you. We have changed this.

  1. Age categories - arbitrary cut-offs. In this case, it would be better to calculate a robust correlation coefficients or examine the difference in mean age between burnout 'categories' (low, moderate, high).

Thank you. We deleted age as it is redundant with years in practice.

  1. I missed a limitation section. There are some further more important psychological and sociodemographic factors which are related to burnout such as social support, coping strategies, or family and financial situation, or work environment, etc. We did not get any information why these determinants was selected in the present study.

We agree. I think the limitations section might have been cutoff in the copy you received as another reviewer noted it stopped mid-sentence. We now include a limitation that we did not assess all relevant factors.

  1. Based on cut-off scores (two levels of burnout)

We deleted this sentence.

  1. I think, it is not interesting. It is very common with using Maslach Burnout Inventory. We suppose that the first indicator of burnout to experience a high level of emotional exhaustion. But a higher level of depersonalization could be critical and it prevent to work. The higher level of emotional exhaustion could be manage for a while by a higher level of personal accomplishment as a coping strategy .

Great point. We have edited in response.

  1. It is a very short conclusion. It is the first mention of resilience trainings, it seems irrelevant in the conclusions. If the aim of study to examine determinants of burnout to develop an effective intervention such as resilience training, please indicate this in the introduction and in the purposes of this study.

Thank you. We have added it to the intro when we clarified our research question.

  1. Please correct the authors' contribution. It is unnecessary to repeat 175-177 for all authors. You can write that all authors approved.......

Thank you. We deleted the redundancy.

Reviewer 2 Report

The presented manuscript has assessed burnout among a group of cardiologists in 3 clinics at Duke University, the whole paper has been written very well, however, there are a few questions/comments that are better answered/be applied.

1-      In the “Participants” sub-title, did the researchers perform this study also for the group of professors (in the cardiology department at the university) too? If the answer is yes, did the researcher performed statistical tests between both clinical and non-clinical categories?

2-      In Abstract; (results) It would be better that the P-value be expressed after each result, and better to explain the meaning of a lower score, and (conclusion) mention the limitation of the study.

3-      In line 95, is there any reference for this category?

Author Response

  1. In the “Participants” sub-title, did the researchers perform this study also for the group of professors (in the cardiology department at the university) too? If the answer is yes, did the researcher performed statistical tests between both clinical and non-clinical categories?

Thank you for asking. The study was just for clinical faculty who saw patients. We subtitle the section Participants as the physicians were our study participants.

  1. In Abstract; (results) It would be better that the P-value be expressed after each result, and better to explain the meaning of a lower score, and (conclusion) mention the limitation of the study.

We now include p-values. Thank you for suggesting this.

  1. In line 95, is there any reference for this category?

Thank you. We now include a reference.

Reviewer 3 Report

SUMMARY OF THE REVIEWER’S REPORT

The Evaluation of the paper titled: “Characteristics associated with burnout among cardiologists in an academic medical setting”

Abstract     
The abstract is loosely written. For example, in the fourth line of the abstract you spoke only about the case study. Moreover, it is not as informative as expected. A standard abstract must present, without leaving any doubt, the objective of the paper precisely; source of data (which is not present in your abstract) and analytical approach used; key findings and any policy implication and recommendations.

Introduction
•     The arguments are fairly presented but the statement that justifies the study does not come clearly (i.e. Why did you started this research?).
•     The introduction does not precisely construct the research problem tackled and does not show how the problem is taken care.
•     The research hypotheses’ are not mentioned in the introduction or clear in the literature review.

Literature review and critical analysis of theories, practices or commentary focusing on existing
documents     
•     The study lacks clear description of the literature review:
•     What I am missing is a description of the review. Did you conduct a systematic literature review? Which years? Key words? What was the literature you found?
•     Can you better describe how you came to your major variables? You have them from the literature review, but how was literature screened to derive these factors.

•    Extend the literature review with papers published in the last two-three years. Fourteen references is too small number for scientific paper;

Methodology and scope of work     
The analytical design is well prepared.

Results
The author has poorly discussed the results of the paper. One would expect to find the previous empirical work enriching the discussions of the results, but unfortunately, that has not been done.

Conclusion and Recommendation     
• The part of the recommendations is rather short, maybe you can strengthen that part in a way which really show the implication of the findings more clearly

Referencing     
Referencing is O.K

Overall, the paper is readable.

Author Response

Abstract     
The abstract is loosely written. For example, in the fourth line of the abstract you spoke only about the case study. Moreover, it is not as informative as expected. A standard abstract must present, without leaving any doubt, the objective of the paper precisely; source of data (which is not present in your abstract) and analytical approach used; key findings and any policy implication and recommendations. 

Thank you for this feedback. We have formalized the abstract.

Introduction
•     The arguments are fairly presented but the statement that justifies the study does not come clearly (i.e. Why did you started this research?).
•     The introduction does not precisely construct the research problem tackled and does not show how the problem is taken care.
•     The research hypotheses’ are not mentioned in the introduction or clear in the literature review.

You are correct. We have reformulated the Intro to clearly state the research questions. We did not have any formal hypotheses, so we cannot state those.

Literature review and critical analysis of theories, practices or commentary focusing on existing
documents     
•     The study lacks clear description of the literature review:
•     What I am missing is a description of the review. Did you conduct a systematic literature review? Which years? Key words? What was the literature you found?
•     Can you better describe how you came to your major variables? You have them from the literature review, but how was literature screened to derive these factors.

  •    Extend the literature review with papers published in the last two-three years. Fourteen references is too small number for scientific paper;

 We added extensively to the literature on burnout in general. There are not many, however, examining cardiologists. We make it clearer that this is why we are conducting this analysis.

Results
The author has poorly discussed the results of the paper. One would expect to find the previous empirical work enriching the discussions of the results, but unfortunately, that has not been done.

We have attempted to elaborate while keeping within the confines of a Brief Report. Again, there is not much literature on predictors of burnout. We attempt to incorporate the previous literature as well as we can.

Conclusion and Recommendation     
• The part of the recommendations is rather short, maybe you can strengthen that part in a way which really show the implication of the findings more clearly

Thank you for this suggestion. We have elaborated on the recommendations.

Reviewer 4 Report

Strengths: Data analytics

Weaknesses: article structure, literature review; the sample; the conclusions

Improvement proposals

Authors should (in the introduction) state the research question, as well as the general and specific objectives of the study.

Line 74 – “We recruited cardiologists to participate in a communication study across 5 outpatient clinics within the Duke Health System. Authors must describe how many participants were invited to participate and what was the criterion for choosing.

Authors must describe the time period of application of the sample.

Row 122 – “Second, female cardiologists had lower mean emotional exhaustion scores than their male counterparts. Authors must place specific values or percentages on the variables studied.

Line 150 – “These results should be interpreted with several noted limitations. The main limitation is that our sample was small and included only. Authors should specify other limitations.

Authors must describe the main conclusions of the study, seeking to respond to the objectives of the investigation. They should also place the value of the study on understanding the topic, limitations and future studies.

Author Response

Authors should (in the introduction) state the research question, as well as the general and specific objectives of the study.

Thank you. We now more clearly state the objectives and research question.

Line 74 – “We recruited cardiologists to participate in a communication study across 5 outpatient clinics within the Duke Health System. Authors must describe how many participants were invited to participate and what was the criterion for choosing. Authors must describe the time period of application of the sample.

You are correct. We now specify both how many we approached and in which time frame.

Row 122 – “Second, female cardiologists had lower mean emotional exhaustion scores than their male counterparts. Authors must place specific values or percentages on the variables studied.

We now list the values as suggested.

Line 150 – “These results should be interpreted with several noted limitations. The main limitation is that our sample was small and included only. Authors should specify other limitations.

Thank you. We now list other limitations.

Authors must describe the main conclusions of the study, seeking to respond to the objectives of the investigation. They should also place the value of the study on understanding the topic, limitations and future studies.

We have strengthened the conclusions of the findings.

Round 2

Reviewer 3 Report

The authors have addressed the point of my concern. I am happy with their corrections. Hence, I would like to recommend this manuscript to be published.

Author Response

Thank you. 

Reviewer 4 Report

Authors should place the "literature review" after the introduction.

They should improve the conclusions, with the considerations observed in this study, limitations and future studies.

Author Response

Authors should place the "literature review" after the introduction.

Thank you. We have added literature review after the introduction.

They should improve the conclusions, with the considerations observed in this study, limitations and future studies.

Thank you for this suggestion. We have elaborated on the conclusions, in light of the limitations.

Round 3

Reviewer 4 Report

Authors should improve the article according to the observations

Author Response

Thank you. We have revised the conclusions to reflect the limitations of the data.